# Constitutive Activation of Guanylate Cyclase by the G86R GCAP1 Variant Is Due to “Locking” Cation-π Interactions that Impair the Activator-to-Inhibitor Structural Transition

**DOI:** 10.3390/ijms21030752

**Published:** 2020-01-23

**Authors:** Seher Abbas, Valerio Marino, Laura Bielefeld, Karl-Wilhelm Koch, Daniele Dell’Orco

**Affiliations:** 1Department of Neuroscience, Division of Biochemistry, University of Oldenburg, 26111 Oldenburg, Germany; seher.abbas@uni-oldenburg.de (S.A.); laura.bielefeld@uni-oldenburg.de (L.B.); karl.w.koch@uni-oldenburg.de (K.-W.K.); 2Department of Neurosciences, Biomedicine and Movement Sciences, Section of Biological Chemistry, University of Verona, 37134 Verona, Italy; valerio.marino@univr.it

**Keywords:** cone dystrophy, neuronal calcium sensor, retinal guanylate cyclase, guanylate cyclase-activating protein 1

## Abstract

Guanylate Cyclase activating protein 1 (GCAP1) mediates the Ca^2+^-dependent regulation of the retinal Guanylate Cyclase (GC) in photoreceptors, acting as a target inhibitor at high [Ca^2+^] and as an activator at low [Ca^2+^]. Recently, a novel missense mutation (G86R) was found in *GUCA1A*, the gene encoding for GCAP1, in patients diagnosed with cone-rod dystrophy. The G86R substitution was found to affect the flexibility of the hinge region connecting the *N*- and *C*-domains of GCAP1, resulting in decreased Ca^2+-^sensitivity and abnormally enhanced affinity for GC. Based on a structural model of GCAP1, here, we tested the hypothesis of a cation-π interaction between the positively charged R86 and the aromatic W94 as the main mechanism underlying the impaired activator-to-inhibitor conformational change. W94 was mutated to F or L, thus, resulting in the double mutants G86R+W94L/F. The double mutants showed minor structural and stability changes with respect to the single G86R mutant, as well as lower affinity for both Mg^2+^ and Ca^2+^, moreover, substitutions of W94 abolished “phase II” in Ca^2+^-titrations followed by intrinsic fluorescence. Interestingly, the presence of an aromatic residue in position 94 significantly increased the aggregation propensity of Ca^2+^-loaded GCAP1 variants. Finally, atomistic simulations of all GCAP1 variants in the presence of Ca^2+^ supported the presence of two cation-π interactions involving R86, which was found to act as a bridge between W94 and W21, thus, locking the hinge region in an activator-like conformation and resulting in the constitutive activation of the target under physiological conditions.

## 1. Introduction

The phototransduction cascade is the first of a complex series of biochemical networks that mediate the conversion of a visual stimulus into a neuronal one. This process is finely tuned by the interconnection of two second messengers, namely, calcium (Ca^2+^) and cyclic guanosine monophosphate (cGMP), whose concentration drops as a consequence of the light-induced activation of the photoreceptor phosphodiesterase 6 [1,2]. The concentration of Ca^2+^ in photoreceptor cells is strictly controlled, ranging from a few hundred nanomolar in the dark to less than 100 nM following photon absorption [3]. Such Ca^2+^-decrease is first detected by Guanylate Cyclase activating protein 1 (GCAP1), an *N*-myristoylated 23 kDa Neuronal Calcium Sensor (NCS) protein [4] responsible for the Ca^2+^/Mg^2+^-dependent regulation of photoreceptor specific Guanylate Cyclase 1 (here and in the following referred to as GC) [5], the enzyme appointed for the cGMP synthesis. In the dark, the Ca^2+^-loaded form of GCAP1 inhibits GC activity, but Ca^2+^-release and Mg^2+^-binding consequential to light-induced intracellular Ca^2+^ drop triggers the conformational switch to the GC-activating form, ultimately resulting in the restoration of the dark cell conditions [6,7].

As a member of the EF-hand calcium-binding protein superfamily, GCAP1 has four EF-hands, three of which are capable of binding Ca^2+^ or Mg^2+^ ions [8] (Figure 1A), arranged in two pairs identifying the *N*- and *C*-domains. These domains are connected structurally by a hinge region, but also functionally by the so-called “myristoyl tug” mechanism [9]. According to this model, Ca^2+^-dissociation from EF4 is coupled with the structural rearrangement of the GC interface encompassing EF1 [10], therefore allowing for the GC inhibitor-to-activator transition of GCAP1.

Mutations in *GUCA1A*, encoding for GCAP1 result in aberrant GC regulation and in retinal degeneration, a set of progressive diseases involving cones, the macula, and in some cases rods, ultimately leading to blindness [11,12,13]. To date, 20 GCAP1 variants have been associated with retinal dystrophy [14,15,16,17,18,19,20,21,22,23,24,25], with one of the most recently discovered being the G86R mutation, located in the previously mentioned hinge region [26]. Biochemical experiments performed with the GC-GCAP1 reconstituted system in the presence of the G86R mutation showed that the complex forms with higher affinity than the WT and it is unable to decelerate the GC at high Ca^2+^-levels corresponding to the dark-adapted state [26]. This mechanism was found to be in line with the clinical cone-rod dystrophy phenotype of the proband and with the autosomal dominant inheritance pattern proven by the segregation study [26]. 

Detailed biochemical and biophysical studies reported a similar common molecular mechanism of dysregulation among the majority of GCAP1 variants [20,27,28,29,30,31,32], namely, a reduced Ca^2+^-affinity due to mutations in EF3 and EF4. On the other hand, some disease-associated mutations resulted in an unaltered Ca^2+^-sensitivity [24,33], suggesting that GC dysregulation may arise from higher affinity for Mg^2+^ or from alterations in the long-range allosteric communication between the EF-hands and the target interface [10].

In this respect, the molecular characterization of the G86R GCAP1 mutant showed features in between the two groups, as the constitutive activation of the GC was due to a concerted effect of: (i) increased affinity for Mg^2+^; (ii) increased affinity for the target; and (iii) lower Ca^2+^-sensitivity. The molecular mechanism underlying these differences was found to reside in the limited flexibility of the hinge region affecting the “myristoyl tug” [26]. Moreover, a close inspection of the three-dimensional structural model of human GCAP1 suggested that the geometry of the hinge region would bring the positively charged guanidino group of R86 sufficiently close to the indole of W94, thus, potentially leading to a stabilizing cation-π interaction (Figure 1A). 

Here, we tested the hypothesis about the formation of a cation-π interaction between R86 and W94 and assessed by in vitro and in silico mutagenesis the role of residue 94 in establishing and maintaining such interaction. We present evidence about the formation, under Ca^2+^ saturating conditions, of bridging lateral cation-π interactions between W94, R86 and W21 responsible for locking the G86R variant in a conformation similar to that of the Mg^2+^-bound, GC-activating WT GCAP1. Such intramolecular lock prevents the physiological activator-to-inhibitor structural transition of G86R GCAP1 that would be expected at physiological levels of Ca^2+^ and is thought to be responsible for the constitutive activation of the GC target under conditions associated with retinal dystrophy.

## 2. Results and Discussion

A visual inspection of the three-dimensional homology model of human GCAP1 (Figure 1A) suggested to us that, following a minor rearrangement of the flexible side chain, the G to R exchange in amino acid position 86 might lead to a stabilizing cation-π interaction involving a W at position 94. We tested this hypothesis by experimental and computational approaches by mutating W to F and L and probing its functional and structural consequences.

### 2.1. GC Regulation by GCAP1 Variants

The functional consequence of the double mutations on the regulation of GC was tested under conditions of low and high Ca^2+^, thus, mimicking, respectively, the light and dark-adapted states of a photoreceptor. Functional assays (Figure 1B) showed that in the absence of Ca^2+^ both G86R and G86R+W94F variants activate GC in a nearly WT-like manner (two-tailed *t*-test, *n* = 3, *p*-value = 0.68 and 0.16, respectively), confirming previous results and proving that purified proteins were functional [26]. On the other hand, G86R+W94L variant greatly affected the maximal activation of the target, resulting in a three-and-a-half-fold decrease of synthesized cGMP in the absence of Ca^2+^ with respect to the WT (20.4 vs. 5.8 nmol cGMP*min^−1^*mg protein^−1^, two-tailed *t*-test, *n* = 3, *p*-value = 0.006). However, both the G86R and the G86R+W94F mutants exhibited high residual GC activity at saturating Ca^2+^-concentrations, indicating constitutive activation of the GC at physiological Ca^2+^, a typical feature of disease causing GCAP1 mutations (see introduction). Indeed, G86R showed a two-fold decrease in cGMP synthesis, G86R+W94F a three-fold decrease and G86R+W94L a four-fold decrease compared to the respective maximal activation, in stark contrast with the behavior of the WT, whose cGMP synthesis in inhibiting conditions was 34-fold lower than that in activating conditions (Figure 1B). 

In conclusion, functional assays suggest that the interaction between R86 and W94 impairs the correct GC-activator-to-inhibitor transition of GCAP1; moreover, such effect seems to be partly rescued when residue 94 is mutated into a less tightly interacting residue.

### 2.2. Mg^2+^/Ca^2+^ Affinity of GCAP1 G86R+W94F/L

In order to understand whether the perturbation observed in the GC catalytic activity exerted by the GCAP1 mutants was due to an alteration of their cation loading state, we investigated the binding of Ca^2+^ and Mg^2+^ to each variant. Ca^2+^ sensor proteins are known to be subjected to differential electrophoretic mobility depending on their Ca^2+^-loading state [34]. GCAP1, as well as other NCS operating in phototransduction, shows a typical mobility shift when the apo-form is compared with the Ca^2+^ or Mg^2+^-loaded forms [24,35], therefore, we monitored the mobility of all GCAP1 variants characterized here in different Ca^2+^/Mg^2+^ loading conditions. In line with previous findings [20,24], all variants displayed lower apparent molecular mass on sodium dodecyl-sulfate polyacrylamide electrophoresis (SDS-PAGE) upon Ca^2+^-binding (Appendix A), independent on the presence of Mg^2+^. The lower apparent molecular mass of the four GCAP1 variants in the presence of Ca^2+^ suggested a similar Ca^2+^-affinity, since the decrease in Ca^2+^-affinity was previously correlated with a less pronounced electrophoretic shift [20].

To investigate in detail ion affinity and Ca^2+^/Mg^2+^-induced conformational changes in the two G86R double variants we performed Isothermal Titration Calorimetry (ITC), of which the results are summarized in Table 1. A particularly thorough procedure of decalcification of both the protein sample and the running buffer permitted the evaluation of a very high affinity site, which was partially occupied under the conditions tested in previous studies [26]. Under these conditions, all Ca^2+^-titrations exhibited both endothermic and exothermic heat response signals (Figure 2). Disease-associated variant G86R displayed a two-fold increase in K_D_ for the two highest-affinity binding sites, and a seven-fold increase in K_D_ for the low-affinity site, with respect to the WT (Table 1). 

Surprisingly, double variant G86R+W94L showed a three-fold increase in the affinity of the highest affinity binding sites compared to the WT, while the opposite trend was observed for the low-affinity site. The concerted presence of G86R and W94F mutation, on the other hand, greatly impacted the Ca^2+^-affinity of all three EF-hands, whose estimated K_D_ resulted to be ranging from three-fold (K_D2_), to ten-fold (K_D3_), up to even two orders of magnitude higher than the WT (K_D1_). 

Overall, the presence of Mg^2+^ lowered the Ca^2+^-affinity of all tested forms, though to a different extent. Indeed, WT showed an almost four-fold reduction in the overall apparent affinity in the presence of Mg^2+^ (K_Dapp_ ~200 nM vs. ~52 nM), while G86R just a one-and-a-half-fold reduction (K_Dapp_ ~243 nM vs. ~167 nM). As to the double mutants, G86R+W94F showed the smallest reduction (K_Dapp_ ~992 nM vs. ~741 nM), while G86R+W94L exhibited the largest decrease in Ca^2+^-affinity, exceeding fourty-fold change (K_Dapp_ ~3.1 µM vs. ~73 nM) (Table 1).

As previously reported by some of us [26], Mg^2+^-binding to WT and G86R GCAP1 showed endothermic heat responses compatible with a two-sites Mg^2+^-binding model and higher Mg^2+^-affinity for G86R compared to WT (K_D1_^G86R^ = 74.5 ± 0.7 µM and K_D2_^G86R^ = 1.75 ± 0.77 µM vs. K_D1_^WT^ = 120 ± 11.3 µM and K_D2_^WT^ = 7.72 ± 1.49 µM; ref. [26]). In this respect, mutations of residue W94 seem to have a detrimental effect on the enhanced Mg^2+^-affinity displayed by the G86R variant, as both K_D1_ and K_D2_ were at least two-fold higher than the WT (Table 1 and Appendix A).

### 2.3. Quaternary Structure and Aggregation Propensity of GCAP1 Variants

NCS proteins can form dimers, and the process may be dynamically influenced by the presence of cations [36]. Since it was recently proven that bovine GCAP1 acts as a functional dimer [37], we tested whether human GCAP1 variants showed similar oligomeric properties under GC activating/inhibiting conditions. To assess the quaternary structure of the GCAP1 variants we employed a combination of dynamic light scattering (DLS) and analytical size exclusion chromatography (SEC). DLS measurements under physiological pH and ionic strength (150 mM) showed high polydispersion (Appendix A), therefore, they did not permit any reliable estimation of the hydrodynamic diameter. Nevertheless, all three GCAP1 variants displayed the co-presence of different oligomeric populations, none of which was clearly prevalent with respect to the others. We then employed analytical SEC to isolate the different populations at lower ionic strength (54 mM), but the chromatograms showed a series of overlapping peaks corresponding to heterogeneous protein populations that could not be clearly resolved, in line with DLS measurements (Appendix A). It should be noticed that, as recently assessed by some of us [38], the estimation of the size of NCS proteins is greatly affected by their hydrophobic properties and by sample homogeneity, resulting in potential misinterpretation of the results. Caution should thus be used when assessing the apparent molecular size and oligomeric states of highly hydrophobic proteins that substantially differ from those used as molecular standards for analytical determinations [38].

The unstable colloidal properties of the GCAP1 variants led us to assess the effects of ion binding on their aggregation propensity under conditions mimicking the physiological ones, by monitoring the time evolution of the count rate of DLS measurements. This index represents the number of photons detected during the measurement time frame and is related to the variation of the quaternary structure of the protein sample, as variations of the count rate indicate the formation or disruption of higher size aggregates. 

While the count rate of apo and Mg^2+^-bound G86R fluctuated around 200–300 kcps during the 2 h of DLS measurements (Figure 3A), the Ca^2+^-loaded form showed a constant increase in the count rate, which ranged from 200 to 2300 kcps in approximately 60 min. The count rate was characterized by broad oscillations. Double mutant G86R+W94F showed a slow increase in count rate (Figure 3B) within 2 h both in the absence of ions (210 to 260 kcps) and upon Mg^2+^-binding (190 to 240 kcps in <1 h). On the other hand, the Ca^2+^-loaded form oscillated around 300 kcps. Interestingly, the double mutant G86R+W94L showed a slight decrease of the count rate (Figure 3C) in the experimental time (∆ count rate <40 kcps) both in the presence of Mg^2+^ and Ca^2+^, while the apo form exhibited an increase in count rate from 250 to 400 kcps and significant oscillations.

Overall, both analytical SEC and DLS results pointed towards an aggregation process of the GC inhibitor state of the pathological GCAP1 variant mediated by the concomitant presence of the G86R mutation and of aromatic residues in position 94. Indeed, both G86R and G86R+W94F displayed an increase in count rate, which was not present in the G86R+W94L double mutant, thus, suggesting that an aromatic side chain is somehow involved in the pathological aggregation process.

### 2.4. Effects of Ca^2+^ Binding on the Intrinsic Trp Fluorescence of GCAP1 Mutants

Binding of Ca^2+^ induces typical conformational changes in GCAP1, which can be followed by monitoring the intrinsic W fluorescence as a function of Ca^2+^ concentration. WT GCAP1 exhibits a biphasic Ca^2+^-dependent change in W fluorescence, showing a decrease in fluorescence below 0.1 µM Ca^2+^ (phase I) and an increase above 0.1 µM Ca^2+^ (phase II). Changes relate to the movement of W residues or conformational changes of its intramolecular vicinity. Indeed, W94 was shown to account for the increase in fluorescence intensity visible in “phase II” of bovine and human GCAP1, while the decrease in fluorescence related to “phase I” was ascribable to the concerted contribution of W21 and W94 of human WT GCAP1. The additional W51 in bovine WT GCAP1 also contributes to phase I. Furthermore, W94 plays an important role in shaping the biphasic conformational transition from the GC-activating to the inhibiting state of both bovine [39,40] and human GCAP1 [26]. Such Ca^2+^-dependent biphasic behavior was still present in the G86R human variant, although shifted towards higher Ca^2+^-range [26] and with lower intensity, and phase I could be eliminated by the presence of saturating Mg^2+^. We investigated the Ca^2+^-dependent behavior of intrinsic W fluorescence GCAP1 in the presence of double mutants substituting the native W94 residue. 

Both double mutants showed a nearly constant decrease in W fluorescence in the range from lower nanomolar concentrations to more than 10 µM Ca^2+^ (Figure 4). The mutants differed in the effect exerted by the presence of Mg^2+^, the W94F mutant showed a slight increase up to 0.1 µM followed by a decrease at higher Ca^2+^-concentrations (Figure 4A). No Mg^2+^-effect was observed for the W94L mutant (Figure 4B). These results confirmed that W94 was the only W residue responsible for phase II, as no increase in fluorescence intensity could be detected even at high Ca^2+^ in either W94L or W94F double variant. Notably, both double variants exhibited conformational changes up to 100 µM Ca^2+^, suggesting a decreased Ca^2+^-sensitivity. However, Mg^2+^ did not suppress phase I in both mutants, which differed from the titration seen with the G86R mutant [26]. This might reflect a lower Mg^2+^-affinity of the double mutants and is consistent with the lower degree of GC activation (Figure 1B).

### 2.5. Effects of Ion Binding on the Conformation and Thermal Stability of GCAP1 Mutants

GCAP1 is a 23 kDa all-α-helix protein consisting of four helix-loop-helix (EF-hand) motifs arranged in two pairs, but only three EF-hands have the consensus sequence for cation binding [41]. Such features allow the assessment of the variations in secondary and tertiary structure by means of circular dichroism (CD) spectroscopy, a technique sensitive to the variations of the microenvironment of aromatic residues in the near UV range (250–320 nm) and to the local folding of the main chain in the far UV range (200–250 nm) [42]. In particular, the ratio in ellipticity at the two minima typical of an α-helix in the far UV region (θ_222_/θ_208_ ) represents a quantitative descriptor for the spectral shape of all-α-helix proteins such as NCS, which can be conveniently used to investigate structural variations upon cation binding [8,43]. Similarly, the Δθ_222_/θ_222_ reports on the increase in ellipticity upon cation binding due to an increase in α helix content, a more compact structure, or both [27].

All three G86R variants showed WT-like θ_222_/θ_208_ values ranging from 0.88 to 0.93, with subtle differences in the spectral shape upon ion binding (Appendix A). In detail, at odds with the WT, G86R exhibited no shape variation upon Mg^2+^-binding and changes in θ_222_/θ_208_ in the presence of Ca^2+^ were less pronounced (0.92 for G86R vs. 0.93 for WT, Table 2). 

On the contrary, both double mutants showed an increase of the ratio already after addition of 1 mM Mg^2+^ (0.90 to 0.92 for G86R+W94F and 0.88 to 0.90 for G86R+W94L, Table 2), suggesting a more significant effect of Mg^2+^ binding on the secondary structure. Interestingly, no effects on the spectral shape could be observed upon Ca^2+^-addition on either mutant.

All GCAP1 mutants showed a three-to-four-fold decrease in Δθ_222_/θ_222_ values upon Mg^2+^ binding with respect to the WT, while Ca^2+^ binding resulted in a 8–10% increase in Δθ_222_/θ_222_ of the G86R variants, more significant than those recorded upon Mg^2+^-binding, yet still smaller than that exhibited by the WT. 

Both G86R and the double mutants showed a flattened near UV CD spectrum in the apo form, compared to the WT (Figure 5), with minor differences mostly ascribable to the F and Y bands. Differences in the F band were expected for the W94F double mutant, not for the G86R and the W94L double mutant, since no additional F residues could affect the signal. The spectra therefore suggest that the G86 to R mutation may somehow affect a hydrophobic patch involving F residues.

Moreover, at odds with the WT and both double mutants, the conformation of the G86R variant resulted to be sensitive to Mg^2+^ as observed by the increased signal mostly in the Y band (Figure 5B–D). Ca^2+^ induced a conformational change in all three variants as clearly shown by the increase of the signal in the Y-W bands and a decrease in the F band, where the signal became more negative. Specifically, mutant G86R and double mutant G86R+W94L showed a WT-like fine structure in the negative F band but a more pronounced positive signal in the Y-W bands, while double mutant G86R+W94F exhibited a positive spectrum in all three aromatic regions, as partly expected by the insertion of an F residue and the simultaneous deletion of the native W.

Thermal denaturation profiles of all three variants (Figure 6) showed a substantial destabilization of the apo form with respect to the WT (7.8 °C < ΔT_m_ < 9.2 °C, Table 2) as well as a mild decrease of the T_m_ in the presence of Mg^2+^ (3.5 °C < ΔT_m_ < 4.5 °C). Nevertheless, the stabilizing effect of Mg^2+^ binding was more pronounced for the G86R single and double variants with respect to the WT (5 °C < ΔT_m_ < 7.2 °C for the variants vs. ΔT_m_ = 0.5 °C for the WT). Finally, no clear transition to an unfolded state was detectable for all three variants upon Ca^2+^ binding (Figure 6), suggesting that the mutations do not significantly affect the stability of the GC-inhibiting forms.

Overall, CD spectroscopy and thermal stability studies confirmed that none of the investigated mutations exerted dramatic effects on the secondary and tertiary structure of GCAP1, neither did they severely compromise the conformational switch triggered by Ca^2+^ or Mg^2+^ under saturating conditions. However, slight but significant spectroscopic differences detected among the single and double mutated variants suggest that the G86R pathogenic variant is characterized by a subtle alteration of key interactions between the newly inserted basic residue and key aromatic amino acids buried in the protein core.

### 2.6. Dynamic Structural Insights on the Interaction between G/R86 and W/F/L94 Provided by Molecular Dynamics Simulations

Since CD spectroscopy proved no major conformational changes in GCAP1 G86R variants, we investigated such subtle variations by running all-atom molecular dynamics (MD) simulations of all variants, including the WT, to provide mechanistic insights on the functional and ion binding differences exhibited by the mutants. We determined the root-mean square fluctuation (RMSF) index over the 300 ns-trajectories, which allows for a quantitative estimation of protein flexibility under the simulated conditions. 

All three variants in the Ca^2+^-loaded form showed overall minor differences in flexibility of the backbone (Figure 7). Notably, G86R+W94L seemed to be overall slightly more rigid than the WT, while the G86R variant showed local higher flexibility in the linker between EF1 and EF2, in all three Ca^2+^-binding loops, in the region of the transient helix and the *C*-terminal helices. Double mutant G86R+W94F, on the other hand, showed an increased flexibility along the *N*-terminal domain, specifically the *N*-term helix and helix αE1. The substantial similarity between RMSF profiles is in line with the thermal denaturation profiles, which were not able to identify a complete folded-unfolded transition in the 20–96 °C range for all tested variants.

Interestingly, RMSF profiles of the EF2/EF3 Mg^2+^-bound GC-activating forms showed more appreciable differences. In detail, GCAP1 WT and double mutants displayed almost overlapping profiles in the entire *N*-terminal lobe up to the hinge region. At odds with the other three variants, the pathological G86R substitution exhibited higher flexibility in the linker region between EF1 and EF2 and strikingly high fluctuations in correspondence of the transient and the *C*-terminal helices (Figure 7B). All three G86R variants displayed lower flexibility in the hinge region compared to the WT, which was propagated throughout the *C*-terminal lobe in the case of G86R+W94F mutant. Finally, G86R+W94L showed higher local flexibility in the region encompassing the transient helix and EF4. In summary, the differences between RMSF profiles of the EF2/EF3-Mg^2+^ form of the simulated variants seem to be in line with thermal denaturation profiles, as none of the two descriptors was substantially altered by the presence of the variants.

In a previous study on the nanosecond protein dynamics of GCAP1 in different cation-loaded states we identified a specific distance (the distance between D168 and R178) as a convenient descriptor of the “twisted accordion” model describing the typical conformational change that GCAP1 undergoes when switching between different signaling states [44]. While MD simulations of WT GCAP1 consistently showed a decrease of D168-R178 distance in the Ca^2+^-loaded state (Figure 8A,B; Appendix A), such distance in Ca^2+^-loaded G86R was almost equal to that measured for EF2/EF3-Mg^2+^ WT and G86R (Figure 8C,D; Appendix A). A close visual inspection of the trajectory of Ca^2+^-loaded G86R variant suggested that the substitution rearranged the sidechains of the proximal W94 and W21 to a conformation compatible with a double cation-π interaction bridged by R86 with 60° < θ < 90 ° and ϕ ~ 0° (where θ is the angle between the aromatic plane and the cation and ϕ is the angle between the cation, the center of the aromatic ring and an aromatic H atom) angles (Figure 8E,G). Despite a suboptimal geometry, such cation-π interaction could stabilize the hinge region acting as a “lock” between the *N*- and *C*-terminal domains. Indeed, relative distances between W21, G/R86 and W94 were smaller for the G86R variant with respect to the WT in both the GC inhibiting (Ca^2+^-loaded) and activating (EF2/EF3-Mg^2+^) forms (Figure 8E,H; Appendix A). Hence, the cation-π interactions mediated by R86 are necessary to bring the indole rings of W94 and W21 significantly close to each other (Figure 8H), thus enhancing the hydrophobic packing of the two domains. Moreover, G86R mutation seemed to affect the variation in G/R86-W21 and W21-W94 distance upon ion exchange, which resulted to be larger for the G86R variant with respect to the WT, while the variation in G/R86-W94 distance highlighted a significant sign inversion between WT and the pathological mutant.

## 3. Materials and Methods

### 3.1. Cloning, Protein Expression and Purification

All *GUCA1A* variants (encoding for GCAP1 G86R, G86R+W94F and G86R+W94F) were generated by PCR site-directed mutagenesis using Q5 High Fidelity kit (New England Biolabs, Frankfurt am Main, Germany) on the cDNA of human GCAP1 in pET11-E6S plasmid (primers available upon request), point mutations were verified by sequencing (GATC Biotech-Germany, Ebersberg, Germany). Human GCAP1 variants were heterologously expressed in BL-21 codon plus *Escherichia coli* cells. Protein purification from the insoluble fraction was achieved by a combination of ammonium sulfate precipitation, size exclusion and anionic exchange chromatography as previously described [20,29,33,45]. Myristoylated forms were obtained with the same protocol after co-transformation with pBB131 containing the gene for yeast (*Saccharomyces cerevisiae*) *N*-myristoyl transferase.

### 3.2. Tryptophan Fluorescence Measurements

Tryptophan fluorescence measurements were performed with a spectrofluorimeter from Photon Technology International (Edison, NJ, USA) as described in ref. [46]. Protein samples were dissolved at a final concentration of 2 µM (measured by Bradford assay [47]) in fluorescence buffer (80 mM 4-(2-hydroxyethyl)-1-piperazineethanesulfonic acid (HEPES) pH 7.5, 40 mM KCl and 1 mM Dithiothreitol (DTT)) in the absence and in the presence of 9 mM MgCl_2_. Spectra were recorded between 300 and 420 nm upon excitation at 280 nm at 25 °C in 500 µL quartz cuvettes. Data reported in Figure 4 refer to the fluorescence emission at 332 nm normalized with respect to the fluorescence intensity at low Ca^2+^ and presented as mean ± standard deviation of two–three independent measurements. Free [Ca^2+^] in protein samples was adjusted using Ca^2+^- ethylene glycol-bis(β-aminoethyl ether)-*N*,*N*,*N*′,*N*′-tetraacetic acid (EGTA) buffer solutions as previously described [48].

### 3.3. Guanylate Cyclase Assay

Recombinant human Guanylate Cyclase 1 (GC, alternatively GC-E, ROS-GC1 or retGC1) was co-expressed with GFP in HEK293 cells as previously detailed [49]. GC activity was measured upon incubation with 10 µM GCAP1 in the presence of 1 mM Mg^2+^ and either 2 mM EGTA or 33 µM free Ca^2+^ as described before [46,48]. Free [Ca^2+^] in protein samples was adjusted using Ca^2+^-EGTA buffer solutions as previously described [48]. Results presented in Figure 1B refer to the average and standard deviation of three independent replicas. The data distributions of the maximal activation of GCAP1 WT and G86R, G86R+W94F and G86R+W94L in the absence of Ca^2+^ passed both the Shapiro-Wilk and equal variance test, therefore were subjected to a two-tailed t-test (passed, *p* = 0.006), where the null hypothesis (equal average between WT and each variant) was rejected only for G86R+W94L (*p* < 0.05). 

### 3.4. Isothermal Titration Calorimetry

Ca^2+^/Mg^2+^ binding to GCAP1 variants was monitored by Isothermal Titration Calorimetry (ITC) using a VP-ITC from MicroCal (Northhampton, MA, USA) as previously detailed [26,50], with a further decalcification step as follows. Purified protein samples were dialyzed overnight against ITC buffer (20 mM Hepes, 60 mM KCl, 4 mM NaCl, pH 7.4) using a 12-14 kDa cutoff membrane. ITC buffer and protein samples were decalcified before measurements (free [Ca^2+^] < 100 nM) using a self-packed gravity flow Chelex 100 column (Bio-Rad, Feldkirchen, Germany). Titrations of 20 µM GCAP1 variants were performed at 25 °C by sequential injections of 5 µL of 0.5 mM CaCl_2_ (in the absence and in the presence of 1 mM MgCl_2_) or 10 mM MgCl_2_ with initial delay of 60 s and 210 s of injection space. For each experiment, three independent repetitions were performed and the reference injection without protein was subtracted from each binding curve. Data was fitted using Origin (MicroCal) to a three-independent binding sites model for Ca^2+^-titrations and to a two-independent binding sites model for Mg^2+^-titrations, allowing the estimation of dissociation constants (K_D_) and enthalpy variations (ΔH).

### 3.5. Analytical Size Exclusion Chromatography

Analytical size exclusion chromatography was performed using the same buffer (30 mM 3-(*N*-morpholino)propanesulfonic acid (MOPS) pH 7.2, 50 mM KCl, 4 mM NaCl and 1 mM DTT) and the same calibration curve as in ref [33] on a BioSep-SEC-S2000 column (Phenomenex, Aschaffenburg, Germany). The molecular weight and Stokes radius of all GCAP1 variants (20 µL injection volume at a concentration of 50 µg/µL) was estimated in the presence of 2 mM Ca^2+^, 3.5 mM Mg^2+^ or 2 mM EGTA, according to refs. [51,52].

### 3.6. Gel Shift Assay

Ca^2+^- and Mg^2+^-dependent protein electrophoresis mobility was determined by using a gel shift assay. Samples consisted in 5 µg protein dissolved in 50 mM Tris/HCl pH 8.0 and incubated for 10 min at RT in a combination of EGTA, Ca^2+^ and Mg^2+^ at a final concentration of 1 mM, then loaded on a 15% SDS-PAGE gel.

### 3.7. Circular Dichroism Spectroscopy and Thermal Denaturation Profiles

GCAP1 variants’ secondary and tertiary structure and thermal denaturation profiles were monitored using a Jasco J-710 (JASCO International, Tokio, Japan) spectropolarimeter and a Peltier type cell holder using the same instrumental setup as previously detailed [43,53]. All CD experiments were performed in 20 mM Tris/HCl pH 7.5, 150 mM KCl, 1 mM DTT buffer where lyophilized proteins were dissolved at a final concentration of 36 and 12 µM (measured by Bradford assay [47]) for near UV (250–320 nm) and far UV (200–250 nm)/thermal denaturation experiments respectively. Near UV and far UV spectra were collected at 37 °C, the first after sequential additions of 500 µM EGTA, 1 mM Mg^2+^ and 1 mM Ca^2+^, the second after sequential additions of 300 µM EGTA, 1 mM Mg^2+^ and 600 µM Ca^2+^. Thermal denaturation profiles were recorded at 222 nm wavelength at a scan speed of 90 °C/h in the 20–96 °C range, sample composition was the same as that for far UV experiments. Melting temperature reported in Table 2 were obtained by fitting raw data to a 4-parameter Hill sigmoid. Reference spectra of the buffer were subtracted from raw data, near UV were normalized by subtracting the average ellipticity in the 310–320 nm range (where no signal should be observed), to avoid artifacts due to cuvette positioning.

### 3.8. Dynamic Light Scattering

Hydrodynamic diameter estimation was performed using a Zetasizer Nano-S (Malvern Instruments, Malvern, England) and polystyrene low volume disposable cuvette using a setup previously optimized by us [54,55]. Lyophilized samples were dissolved in 20 mM Tris-HCl, 150 mM KCl, 1 mM DTT and either 500 µM EGTA, 500 µM EGTA and 1 mM Mg^2+^ or 500 µM EGTA, 1 mM Mg^2+^ and 1 mM Ca^2+^ were added. Samples were centrifuged for 30 min at 4 °C and 18,000 × G and filtered and through an Anotop 10 (Whatman, Maidstone, England) filter with 20 nm cutoff, then equilibrated for 3 min at 25 °C before starting the >60 measurements, each consisting of 12 repetitions. 

### 3.9. Molecular Dynamics Simulations

The homology model of myristoylated human GCAP1 was built using as a template the three-dimensional structure of the Ca^2+^-loaded chicken GCAP1 [41] as previously elucidated [27]. Mg^2+^-bound states were modeled by removing or replacing Ca^2+^ ions with Mg^2+^ ions from the different EF-hands as explained in ref. [8]. In silico mutagenesis of all variants was performed using the aforementioned structure as a template and the highest-scored rotamer suggested by the “Mutate Residue” function provided by Maestro v. 12.2.012 (Schrodinger) software. MD simulations and analyses were performed on GROMACS v. 2016.1 simulation package [56] employing a variant of CHARMM36m [57] all-atom force field comprehensive of the parameters of the myristoylated Gly (available upon request). The pipeline for energy minimization, system equilibration and production was the same as previously described [10,58] with the only difference consisting in the production phase, in which the simulated time frame was 300 ns. The trajectories were then subjected to distance and RMSF analysis computed by the gmx rmsf, gmx distance and gmx gangle functions implemented in GROMACS. Smoothing of the distance plots shown in Figure 8 was achieved calculating the running average and standard deviation over 1 ns (100 points) of the raw distance values.

## 4. Conclusions

In conclusion, our data collectively suggest that the residual GC activity induced by the cone-rod dystrophy-associated G86R GCAP1 mutant under high Ca^2+^ may be caused by the intramolecular lock constituted by the R86-W21 and R86-W94 cation-π interactions, which ultimately prevent the correct structural rearrangement necessary for promoting the activator to inhibitor conformational transition.

## Figures and Tables

**Figure 1 ijms-21-00752-f001:**
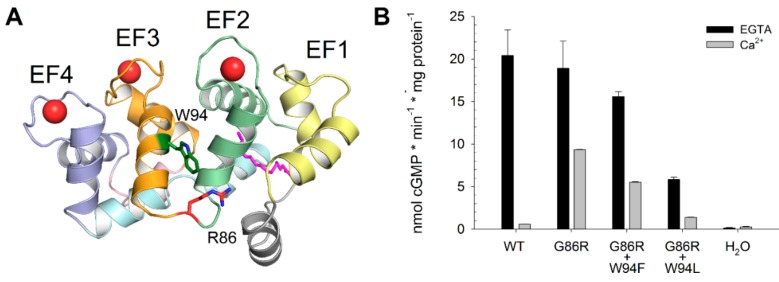
Three-dimensional structure of Ca^2+^-loaded Guanylate Cyclase activating protein 1 (GCAP1) G86R (**A**) and Guanylate Cyclase (GC) regulation by GCAP1 WT, G86R, G86R+W94F and G86R+W94L variants expressed in nmol cGMP * min^−1^ * mg protein^−1^ (**B**). (**A**) Protein structure is represented as a cartoon, Ca^2+^ ions are displayed as red spheres, the myristoyl group, R86 and W94 are shown as sticks and colored magenta, red and green, respectively, with N atoms colored blue. *C*-terminal helix is depicted in grey, EF1 in yellow, EF2 in green, EF3 in orange, EF4 in blue and *N*-terminal helices in cyan. (**B**) Cell membranes containing GC were incubated with ~10 µM GCAP1 variants in the presence of 1 mM Mg^2+^ and 2 mM ethylene glycol-bis(β-aminoethyl ether)-*N*,*N*,*N*′,*N*′-tetraacetic acid (EGTA) (black) or 33 µM free Ca^2+^ (grey), H_2_O was used as control. Data refer to the average and standard deviation of three independent replicas.

**Figure 2 ijms-21-00752-f002:**
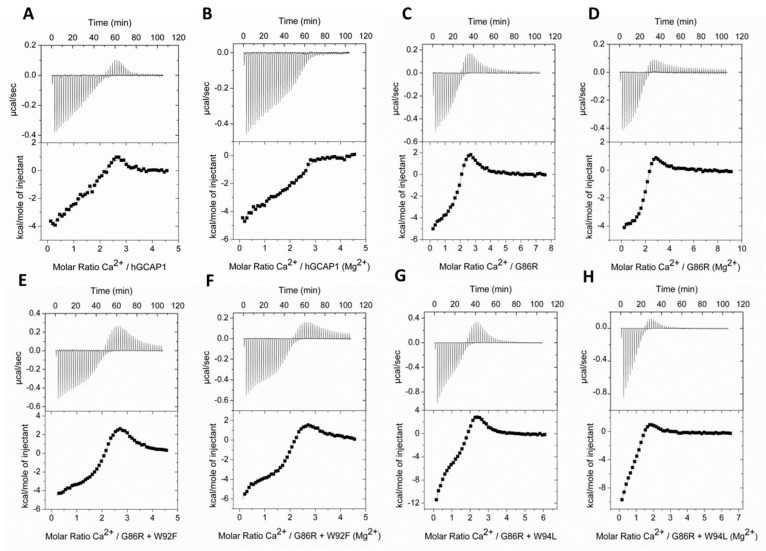
Ion binding to GCAP1 variants monitored by Isothermal Titration Calorimetry (ITC). Representative Ca^2+^ titrations of 20 µM GCAP1 variants in the absence (panels **A**, **C**, **E** and **G**) and in the presence of 1 mM Mg^2+^ (panels **B**, **D**, **F** and **H**). (Representative Mg^2+^-titrations of 20 µM GCAP1 G86R+W94F and G86R+W94L are shown as Appendix A). Panels represent heat pulses and molar enthalpy changes corresponding to each injection. Ca^2+^-titration data was fitted to a three-sites binding model, while Mg^2+^-titration data was fitted to a two-sites binding model yielding macroscopic dissociation constants (K_D_) and enthalpy changes (ΔH) for each binding site, listed in Table 1.

**Figure 3 ijms-21-00752-f003:**
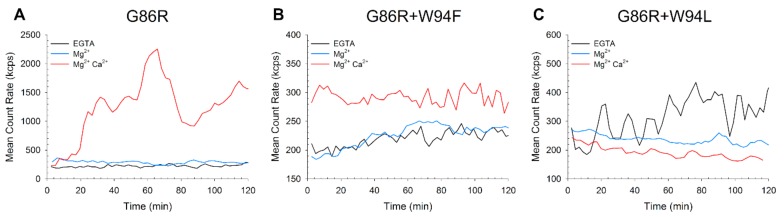
Time evolution of the mean count rate of GCAP1 variants. The mean count rate of GCAP1 variants G86R (**A**), G86R+W94F (**B**), G86R+W94L (**C**) was monitored at 37 °C collecting measurements every ~2 min for 2 h, where each measurement consisted of 12–15 runs. Samples consisted in ~30 µM protein dissolved in 20 mM tris(hydroxymethyl)aminomethane (Tris)-HCl, 150 mM KCl, 1 mM Dithiothreitol (DTT) in the presence of 500 µM EGTA (black), 500 µM EGTA and 1 mM Mg^2+^ (blue) or 500 µM EGTA, 1 mM Mg^2+^ and 1 mM Ca^2+^ (red).

**Figure 4 ijms-21-00752-f004:**
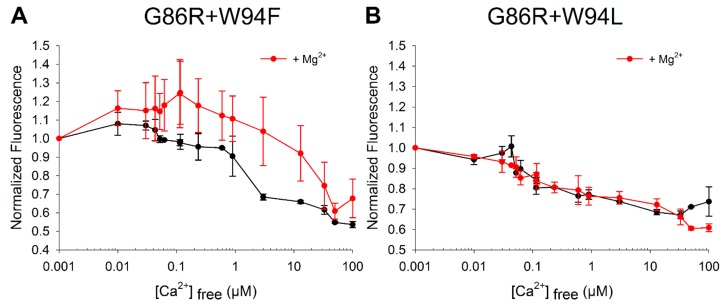
Intrinsic fluorescence emission of GCAP1 mutants G86R+W94F (**A**) and G86R+W94L (**B**) upon Ca^2+^ titration. Samples were dissolved in 80 mM 4-(2-hydroxyethyl)-1-piperazineethanesulfonic acid (HEPES) pH 7.5, 40 mM KCl and 1 mM DTT buffer to a final concentration of 2 µM in the absence (black) and in the presence of 9 mM Mg^2+^ (red). The Ca^2+^-dependent fluorescence emission was monitored at 332 nm upon excitation at 280 nm.

**Figure 5 ijms-21-00752-f005:**
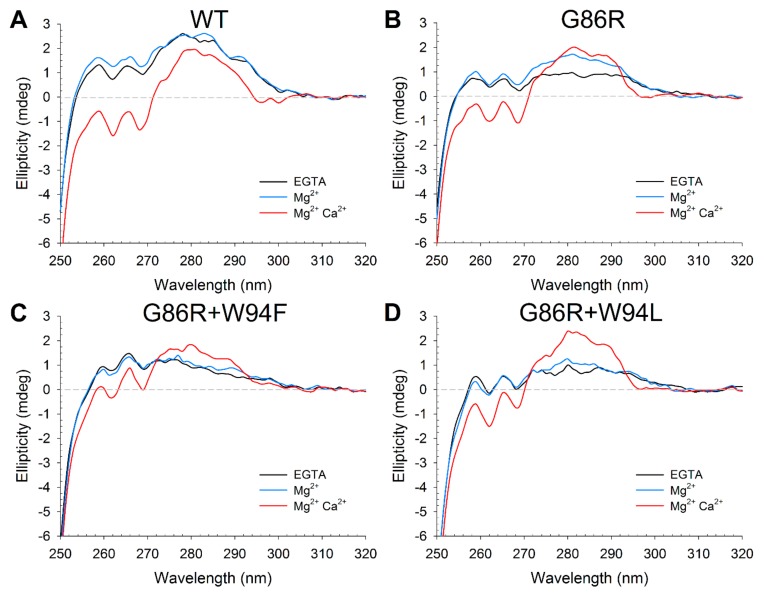
Tertiary structure changes occurring in GCAP1 variants upon ion binding, monitored by Circular Dichroism (CD) spectroscopy. Near Ultraviolet (UV) CD spectra of ~36 µM GCAP1 WT (**A**), G86R (**B**), G86R+W94F (**C**) and G86R+W94L (**D**) in the presence of 500 µM EGTA (black), 500 µM EGTA and 1 mM Mg^2+^ (blue) or 500 µM EGTA, 1 mM Mg^2+^ and 1 mM µM Ca^2+^ (red). All experiments were performed at 37 °C in 20 mM Tris-HCl, 150 mM KCl, 1 mM DTT buffer.

**Figure 6 ijms-21-00752-f006:**
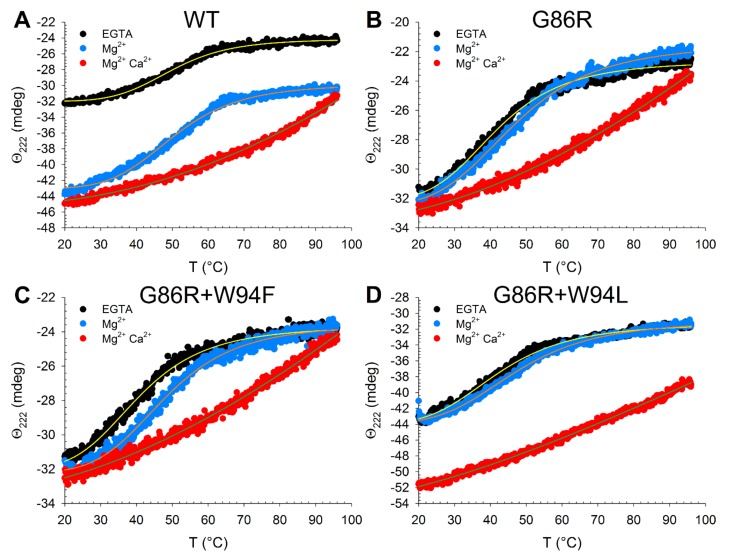
Thermal stability of GCAP1 variants monitored by far UV CD spectroscopy. Thermal denaturation profiles of ~12 µM GCAP1 WT (**A**), G86R (**B**), G86R+W94F (**C**) and G86R+W94L (**D**) in the presence of 300 µM EGTA (black), 300 µM EGTA and 1 mM Mg^2+^ (blue) or 300 µM EGTA, 1 mM Mg^2+^ and 600 µM Ca^2+^ (red). The profiles were recorded following the ellipticity at 222 in the 20–96 °C temperature range. All experiments were performed in 20 mM Tris-HCl, 150 mM KCl, 1 mM DTT buffer, data was fitted to a four-parameter Hill sigmoid, whose estimated T_m_ is reported in Table 2.

**Figure 7 ijms-21-00752-f007:**
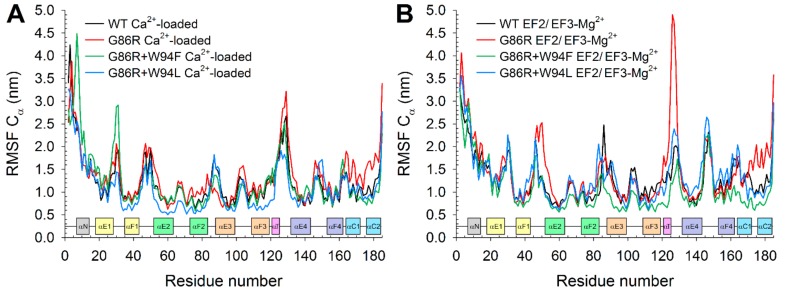
Root-mean square fluctuation calculated over 300 ns MD simulations on Cα of GCAP1 WT (black), G86R (red), G86R+W94F (green) and G86R+W94L (blue) in their Ca^2+^-loaded (**A**) and EF2/EF3-Mg^2+^ (**B**) forms. Insets show protein secondary structure colored according to Figure 1A.

**Figure 8 ijms-21-00752-f008:**
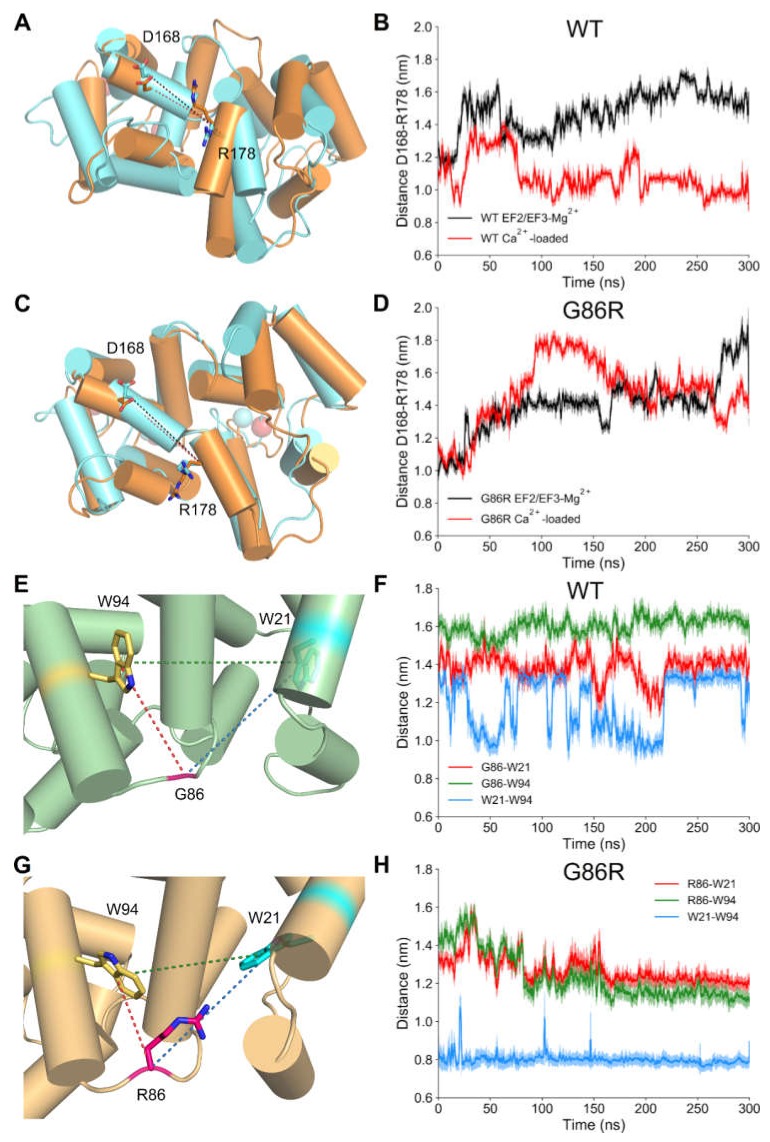
Structural descriptors of GCAP1 WT and G86R calculated on the 300 ns MD trajectories. The three-dimensional structure of EF2/EF3-Mg^2+^ and Ca^2+^-loaded GCAP1 WT (**A**) and G86R (**C**) is shown in cartoon and colored in cyan and orange, respectively, residues D168 and R178 are shown in sticks, O and N atoms are colored in red and blue, respectively, Ca^2+^ ions are shown as red spheres, Mg^2+^ ions are shown as green spheres. Time evolution of the distance between Cα of residues D168 and R178 of GCAP1 WT (**B**) and G86R (**D**) in their EF2/EF3-Mg^2+^ (black) and Ca^2+^-loaded (red) forms. Structural detail of the distance between residues G/R86 and W21 and W94 of GCAP1 WT (**E**) and G86R (**G**) in their Ca^2+^-loaded forms; GCAP1 WT and the G86R three-dimensional structure is shown as cartoons and colored in green and orange, respectively. Residues W21, G/R86 and W94 are shown as sticks and colored in cyan, purple and yellow respectively, O atoms are displayed in red, N atoms are displayed in blue. G/R86-W21 distance is shown as a dashed red line, G/R86-W94 distance is shown as a dashed green line, W21-W94 distance is shown as a dashed in blue line. Time evolution of the distance between residues G/R86 and, W21 and W94 of GCAP1 WT (**F**) and G86R (**H**) in their Ca^2+^-loaded forms. G/R86-W21 distance is shown in red, G/R86-W94 distance is shown in green, W21-W94 distance is shown in blue, light colors represent standard deviation, details about the smoothing procedure are reported in the methods section.

**Table 1 ijms-21-00752-t001:** Thermodynamic parameters of Ca^2+^ and Mg^2+^ binding to GCAP1 variants estimated by Isothermal Titration Calorimetry (ITC) measurements. Ca^2+^-titrations were performed in the absence (upper half) or in the presence of 1 mM Mg^2+^ (lower half). Dissociation constants and enthalpy changes ΔH are reported as average ± standard deviation of 3-4 repetitions after fitting Ca^2+^-titration data to a 3-sites binding model and Mg^2+^-titration data to a 2-sites binding model.

GCAP1 Variant	GCAP1 Ca^2+^ Titration (Three Site Model)					
	Apparent Dissociation Constant K_D_			Enthalpy Change ΔH (kcal/mol)		
	K_D1_ (nM)	K_D2_ (nM)	K_D3_ (µM)	ΔH_1_	ΔH_2_	ΔH_3_
WT	16.3 ± 1.2	36 ± 8	0.25 ± 0.05	−4.2 ± 0.2	−5 ± 4	1.20 ± 0.05
G86R	33 ± 2	78 ± 12	1.80 ± 0.05	−5.2 ± 0.1	−2.2 ± 0.1	3.20 ± 0.08
G86R+W94L	6 ± 4	7 ± 4	0.82 ± 0.13	−12 ± 2	3.5 ± 0.5	2.6 ± 0.2
G86R+W94F	1.3 ± 0.8 (µM)	101 ± 23	3.1 ± 0.3	−7.4 ± 1.2	−1.2 ± 0.2	5.4 ± 0.2
	**+1 mM Mg^2+^**					
WT	77 ± 28	129 ± 17	0.8 ± 0.2	−3.5 ± 0.4	−1.2 ± 0.5	0.10 ± 0.01
G86R	62 ± 27	145 ± 12	1.6 ± 1.1	−4.0 ± 0.1	−3.7 ± 0.1	0.20 ± 0.02
G86R+W94L	138 ± 12	628 ± 30	358.0 ± 0.4	−11.0 ± 0.1	3.8 ± 0.4	−6 ± 3
G86R+W94F	1.9 ± 1.2 (µM)	133 ± 34	4.6 ± 1.8	−9.2 ± 2.0	−2.6 ± 0.5	3.6 ± 0.3
**GCAP1 Variant**	**GCAP1 Mg^2+^ Titration (Two Site Model)**					
	**K_D1_ (µM)**	**K_D2_ (µM)**		**ΔH_1_**		**ΔH_2_**
G86R+W94L	648 ± 144	15 ± 8		0.34 ± 0.08		21 ± 6
G86R+W94F	238 ± 123	112 ± 41		1.4 ± 0.5		12.3 ± 1.4

GCAP1—Guanylate Cyclase activating protein 1; WT—Wild Type; K_D1_, K_D2_, K_D3_—Apparent dissociation constant of binding sites 1, 2 and 3; ΔH_1_, ΔH_2_, ΔH_3_—Enthalpy change of calcium/magnesium binding to sites 1, 2 and 3; G86R—Gly 86 to Trp 86 variant; W94L—Trp 94 to Leu 94 variant; W94F—Trp 94 to Phe 94 variant.

**Table 2 ijms-21-00752-t002:** Spectral shape descriptors and melting temperatures of GCAP1 WT, G86R, G86R+W94F and G86R+W94L obtained by Circular Dichroism (CD) spectroscopy upon ion binding. Δθ_222_/θ_222_ was calculated as (θ_222_^ion^-θ_222_^EGTA^)/θ_222_^EGTA^, T_m_ estimation was achieved by fitting thermal denaturation profiles to a 4-parameter Hill sigmoid.

GCAP1 Variant	Experimental Condition	θ_222_/θ_208_	Δθ_222_/θ_222_	T_m_ (°C)
WT	EGTA	0.88	-	49.8
	Mg^2+^	0.90	0.08	50.3
	Mg^2+^ Ca^2+^	0.93	0.15	>96
G86R	EGTA	0.88	-	41.6
	Mg^2+^	0.88	0.03	46.8
	Mg^2+^ Ca^2+^	0.92	0.10	>96
G86R+W94F	EGTA	0.90	-	40.6
	Mg^2+^	0.92	0.02	47.8
	Mg^2+^ Ca^2+^	0.92	0.08	>96
G86R+W94L	EGTA	0.88	-	42
	Mg^2+^	0.90	0.02	47
	Mg^2+^ Ca^2+^	0.90	0.09	>96

EGTA—ethylene glycol-bis(β-aminoethyl ether)-*N*,*N*,*N*′,*N*′-tetraacetic acid.

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
