# Peer review of "Constitutive Activation of Guanylate Cyclase by the G86R GCAP1 Variant Is Due to “Locking” Cation-π Interactions that Impair the Activator-to-Inhibitor Structural Transition"

_ijms, 2020, doi:10.3390/ijms21030752_

Round 1

Reviewer 1 Report

The authors defined the role of residue 94 in establishing and maintaining such interaction of a cation-π interaction between R86 and W94. The experimental design was well organized and the results were supported by the data. However, the discussion is still poor to understand the clinical significance of the GCAP1 variants.

The authors described that twenty GCAP1 variants have been associated with retinal dystrophy and some disease-associated mutations resulted in an unaltered Ca2+-sensitivity [24,33]. However there is no clear explanation of the pathological relevance of missense mutation (G86R) in the discussion.

Likewise, pathophysiological relevance of he formation of a cation-π interaction between R86 and W94.

In another words, there are no description of the mutation relevance in patients diagnosed with cone-rod dystrophy in the discussion, please add the missing information.

Author Response

The authors defined the role of residue 94 in establishing and maintaining such interaction of a cation-π interaction between R86 and W94. The experimental design was well organized and the results were supported by the data.

Response: We thank the Reviewer for the positive comments.

However, the discussion is still poor to understand the clinical significance of the GCAP1 variants. The authors described that twenty GCAP1 variants have been associated with retinal dystrophy and some disease-associated mutations resulted in an unaltered Ca2+-sensitivity [24,33]. However there is no clear explanation of the pathological relevance of missense mutation (G86R) in the discussion.

Response: The goal of this work was not to highlight the pathological relevance of the G86R mutation in GCAP1, as this was already done by some of us in a previous work (ref. 26). However, we agree with the Reviewer that a reference to the effects of the mutation on the dysregulation of the guanylate cyclase and to the clinical phenotype was missing in the original submission. In the revised version we have added a full paragraph on that  (lines 67-72).

Likewise, pathophysiological relevance of he formation of a cation-π interaction between R86 and W94.

Response: We think that our work clearly discusses the hypothesis that the dysregulation of the GC investigated at the molecular level can be attributed to  cation-π interactions in the G86R GCAP1 variant. This is the link between molecular interpretation and a pathophysiological state described earlier (Ref. 26)

In another words, there are no description of the mutation relevance in patients diagnosed with cone-rod dystrophy in the discussion, please add the missing information.

Response:Please see the response to the previous points.

Reviewer 2 Report

The current paper is an interesting contribution to the field of cone-rod degeneration as it shed more lights on how G86R (a gof mutation) in GCAP1 alters the regulation of retinal GC and causes the retinal dysrophy. Once the below comments get included, I highly recommend the acceptance of the article.

Major comments:

Line 90: It is not familiar to merge results and discussion together, authors could separate these two sections.

Line 112:

When authors have compared G86R+W94L variant to WT in terms of GC regulation they showed P values without describing any statistical analysis part in methods section. Authors are encouraged to add a statistical analysis section and update the legend section of figure 1B with the obtained significance.

Minor comments:

Lines 63, 387: Genes symbols need to be italicized  

Line 281: Add the acronym for CD in legend of figure 5.

Author Response

The current paper is an interesting contribution to the field of cone-rod degeneration as it shed more lights on how G86R (a gof mutation) in GCAP1 alters the regulation of retinal GC and causes the retinal dysrophy. Once the below comments get included, I highly recommend the acceptance of the article.

Response: We thank the Reviewer for the positive comments.

Major comments: Line 90: It is not familiar to merge results and discussion together, authors could separate these two sections.

Response: We believe that, in this type of manuscript discussing the implications of the results right after their presentation allows a deeper general understanding, and it is better for the flow of information. The concluding paragraph then effectively summarizes the general finding. Moreover, it is explicitly allowed by IJMS guidelines to merge the Results and Discussion session, therefore we prefer to leave the manuscript structure as is.

Line 112: When authors have compared G86R+W94L variant to WT in terms of GC regulation they showed P values without describing any statistical analysis part in methods section. Authors are encouraged to add a statistical analysis section and update the legend section of figure 1B with the obtained significance.

Response: We agree that some further detail on the statistical analysis was necessary. In the revised manuscript we have explained in paragraph 2.1 the statistical results in deeper detail, and we have added (lines 416-421) a paragraph on the tests that were performed, modifying the legend as well to Figure 1B.

Minor comments:

Lines 63, 387: Genes symbols need to be italicized  

Line 281: Add the acronym for CD in legend of figure 5.

Response: Thank you for the suggestion, this has been done.